# Brain connectivity-guided, Optimised theta burst transcranial magnetic stimulation to improve Central Pain Modulation in knee Osteoarthritis Pain (BoostCPM): protocol of a pilot randomised clinical trial in a secondary care setting in the UK

Marianne Drabek [ID] ,[1,2,3] Duncan Hodkinson,[1,2,3] Suzanne Horvath,[2] Bonnie Millar,[2,3,4] Stefan Pszczolkowski Parraguez,[1,2] Christopher R Tench [ID] ,[2] Radu Tanasescu,[2,5] Sudheer Lankappa,[6] Richard Morriss [ID] ,[1,2] David Walsh,[2,3,4] Dorothee P Auer[1,2,3]

For numbered affiliations see end of article.

**Correspondence to**
Professor Dorothee P Auer;
dorothee.auer@nottingham.ac.uk

## ABSTRACT

**Introduction** Chronic pain is a common health problem that is not efficiently managed by standard analgesic treatments. There is evidence that treatment resistance may result from maladaptive brain changes in areas that are fundamental to the perception of pain. Knee osteoarthritis is one of the most prevalent causes of chronic pain and commonly associated with negative affect. Chronic knee osteoarthritis pain is also associated with altered right anterior insula functional connectivity. We posit that reversal of these brain circuit alterations may be critical to alleviate chronic pain and associated negative affect, and that this can be achieved through non-invasive neuromodulation techniques. Despite growing interest in non-invasive neuromodulation for pain relief and proven efficacy in depression, results in chronic pain are mixed with limited high-quality evidence for clinical and mechanistic efficacy. Limitations include patient heterogeneity, imprecision of target selection, uncertain blinding and protocols that may deliver pulses at subclinical efficacy.

**Methods and analysis** We hence developed an optimised treatment protocol of connectivity-guided intermittent theta-burst stimulation (iTBS) targeting the left dorsolateral prefrontal cortex with accelerated delivery on four consecutive days (allowing 4 days within the same week as protocol variation) with five daily treatment sessions that will be piloted in a sham-controlled design in 45 participants with chronic knee pain. This pilot study protocol will assess feasibility, tolerability and explore mechanistic efficacy through serial functional/structural magnetic resonance imaging (MRI) and quantitative sensory testing.

**Ethics and dissemination** This pilot trial has been approved by the Ethics Committee Cornwall and Plymouth.

## STRENGTHS AND LIMITATIONS OF THIS STUDY

⇒ The protocol combines connectivity-guided neuronavigation for improved precision to target the prefronto-insular circuit with accelerated delivery of five daily intermittent theta-burst stimulation sessions over 4 days in 1 week for enhanced efficacy and efficiency in a sham-controlled randomized controlled trial (RCT) design.

⇒ Advanced serial MRI of brain network function complements psychophysical and questionnaire outcome markers before and after transcranial magnetic stimulation protocol to explore mechanistic efficacy.

⇒ Recruitment is predominantly from a research-ready community cohort which together with the single centre design may limit generalisability.

⇒ The intense protocol may only be accommodated and tolerated by a subgroup of pain patients.

⇒ The total number of 36 000 pulses and the permitted reduction of resting motor threshold below 80% may limit efficacy of the intervention.

Results of the pilot trial will be submitted to peer-reviewed journals, presented at research conferences and may be shared with participants and PPI/E advisors.
**Trial registration number** ISRCTN15404076.

## INTRODUCTION

Chronic pain is a major burden for society, economy, health systems and individuals. There is a lack of effective analgesic treatments in chronic pain with an urgent need for novel treatments. Maladaptive brain mechanisms are increasingly recognised as

key contributors to the development and persistence of pain.[1] A feature is the altered communication between brain areas that are involved in the processing of pain signals and the modulation of the associated pain perception, demonstrated by dysfunctional connectivity of the right anterior insula (rAI).[2]

These maladaptive functional brain network changes are thought (i) to drive the experience of chronic pain and central pain sensitisation, (ii) to be potentially reversible by non-invasive neuromodulation, and thus support the notion that interventions that normalise the brain's (and in particular fronto-limbic) connectivity will also reverse central pain augmentation or associated affective experience.

Repetitive transcranial magnetic stimulation (rTMS) is a promising tool to non-invasively and focally stimulate the brain through induction of small electrical currents. Choice of stimulation protocols are thought to determine excitatory or inhibitory local and downstream effects with acute and longer-term neuromodulatory effects on targeted brain circuits. The exact mechanism of acute neuromodulatory and neuroplastic effects and the substantial interindividual and intraindividual variability are poorly understood.[3] Clinical efficacy of rTMS is well-established for the treatment of major depression, approved by the Federal Drug Administration (FDA) in the USA and the National Institute of Clinical Excellence in the UK. In chronic pain, to date evidence for effective neuromodulation is limited to invasive protocols and selected indications.[4] There is however considerable recent interest in non-invasive brain stimulation (NIBS) trials in chronic pain yet results to date are mixed. Meta-analyses and systematic reviews have described NIBS as having 'poor or inconclusive' analgesic effects across sites and pain disorders,[4–6] 'minimally clinically relevant',[6] to 'promising'[7] and 'probable efficacy' for either motor cortex (M1) or dorsolateral prefrontal cortex (DLPFC) stimulation.[8] There are major quality concerns with most studies considered poor[4] due to underpowered sample sizes, lack of blinding, inadequate randomisation or missing sham for comparisons. Moreover, there are uncertainties regarding the most effective brain stimulation site, type of stimulation, target population. Nevertheless, pain relief was frequently found to be >30%[9] which encourages to seek further improvements and address limitations in order to fully assess and exploit the potential that this technology may hold for pain treatment.

To address these shortcomings we chose to target the left DLPFC as a particularly promising yet relatively understudied target site for rTMS in chronic pain due to its strong modulatory influence on the rAI, a key hub in pain processing with known altered connectivity in chronic pain.[2] Stimulating the left DLPFC is expected to modulate the prefronto-insular pathway as a putative central pain modulatory effect. This pathway is also a key target in the clinically effective rTMS treatment of major depression suggesting a second putative beneficial effect through reduction of negative affect and related pain augmentation.

For target selection, we chose resting state functional MRI (rsfMRI) based effective connectivity to determine the coordinates within the anatomically defined left DLPFC that show the strongest effective connectivity to the rAI as in our previous work.[10 11] In addition, recently accelerated, optimised TMS protocols were reported to be safe and similarly effective to standard rTMS using intermittent theta burst stimulation (iTBS).[12–15] There are several advantages expected from iTBS protocols that are faster than standard rTMS and may have stronger neuroplasticity effects. Also, the multiple treatment blocks per day with 50 min intervals are expected to further enhance effectiveness and allow to deliver a higher cumulative weekly dose with the benefit of shorter overall treatment course.

Motivated by the promising results of the optimised iTBS protocol in major depression, we translated and adapted the approach to chronic musculoskeletal pain, which, if effective, would greatly improve clinical interest in this technique both for cost and practical reasons. As study population we chose knee osteoarthritis (OA) as one of the most common causes of chronic musculoskeletal pain and to avoid potential heterogeneity from different pain aetiology. Based on the known comorbidity, we anticipate that the study population of moderate chronic knee OA pain will report higher negative affect than the general population which may further increase the likelihood of neuromodulatory efficacy at the chosen target site.

This parallel, sham-controlled pilot study will investigate a 4-day (over 1 week) intense, imaging-guided iTBS protocol that is individually targeted within the left DLPFC to maximise the connectivity with the rAI. Study participants with moderately severe, chronic knee pain will undergo serial MRI, behavioural and quantitative sensory testing (QST) before and after the iTBS/sham intervention and online recording of patient-reported outcomes until week 18 to guide the protocol and sample size for future RCT interventions.

### Objectives

1. To test the feasibility and tolerability of an optimised accelerated 4-day (over 1 week) iTBS protocol with five daily sessions (total pulses 36 000 aimed at 80% resting motor threshold (RMT)) using connectivity-guided targeting of the left DLPFC in chronic knee OA patients with moderately severe pain.
2. To assess the variance of mechanistic outcome markers comparing the intense iTBS protocol vs sham.
3. To explore clinical efficacy metrics, and the feasibility of online recorded patient-reported outcome measures (PROMs) in the iTBS and sham arm.
4. To explore the effects of iTBS on negative affect, stress and arousal between iTBS and sham.
5. To assess feasibility of blinding using sham TMS.

## METHODS AND ANALYSIS

### Trial design

Single centre, sham controlled randomised mechanistic pilot study at the University of Nottingham with a parallel group study design, whereby participants will either receive active iTBS or sham iTBS with random allocation of 2:1. The uneven allocation was chosen to achieve blinding but preferencing power for statistical analyses in the active condition.

### Hypotheses

#### Primary mechanistic hypotheses

1. Accelerated, personalised iTBS of the DLPFC modulates *functional activity and connectivity* of brain circuits involved in central pain control in chronic musculoskeletal pain compared with sham controls. The direction of change is expected to reduce hallmark features of the pain connectome, such as reduction of the abnormally increased rAI-posterior default mode network anticorrelation.
2. Accelerated, personalised iTBS of the DLPFC increases endogenous antinociception in chronic musculoskeletal pain compared with sham controls.

#### Secondary hypotheses

1. Accelerated, personalised iTBS of the DLPFC reduces pain catastrophising, anxiety and improves mood in chronic musculoskeletal pain compared with sham controls.
2. Accelerated, personalised iTBS of the DLPFC reduces sensitivity to experimental pain at the end of the intervention compared with sham controls.

3. Accelerated, personalised iTBS of the DLPFC reduces physiological arousal and reported stress levels compared with sham controls.

### Study procedures

Participants will first be assessed over phone for meeting safety and eligibility criteria before giving informed written consent. Participants will be asked to attend for baseline assessments (visit 1) (ca. 2 hours including MRI, and psycho-physical assessment including brief experimental pain exposure) (see figure 1). Then participants will undergo an intervention week with four visits (visit 2–5) lasting about half a day each. Each visit will consist of five blocks of iTBS or sham with inter-block interval of ~50 min during which participants will be offered refreshments and food in a comfortable waiting area before getting prepared for the next session. The sixth and final study visit is a repeat of the baseline visit, and will be scheduled within 18–72 hours of the last treatment session. This will be followed by online patient self-recording of pain and emotional outcomes up to 16 weeks.

### Recruitment and inclusion criteria

Participants will be recruited primarily from previous participants who consented to be contacted (such as the community-based Investigating Musculoskeletal Health and Wellbeing cohort)[16] and additional recruitment pathways via posters, general practitioner surgeries and other NHS services (joint replacement waiting list) or corecruited from ongoing studies that allow corecruitment as required. Individuals who consented to be contacted about new studies, will be contacted by the study team who originally consented the participant; only if the

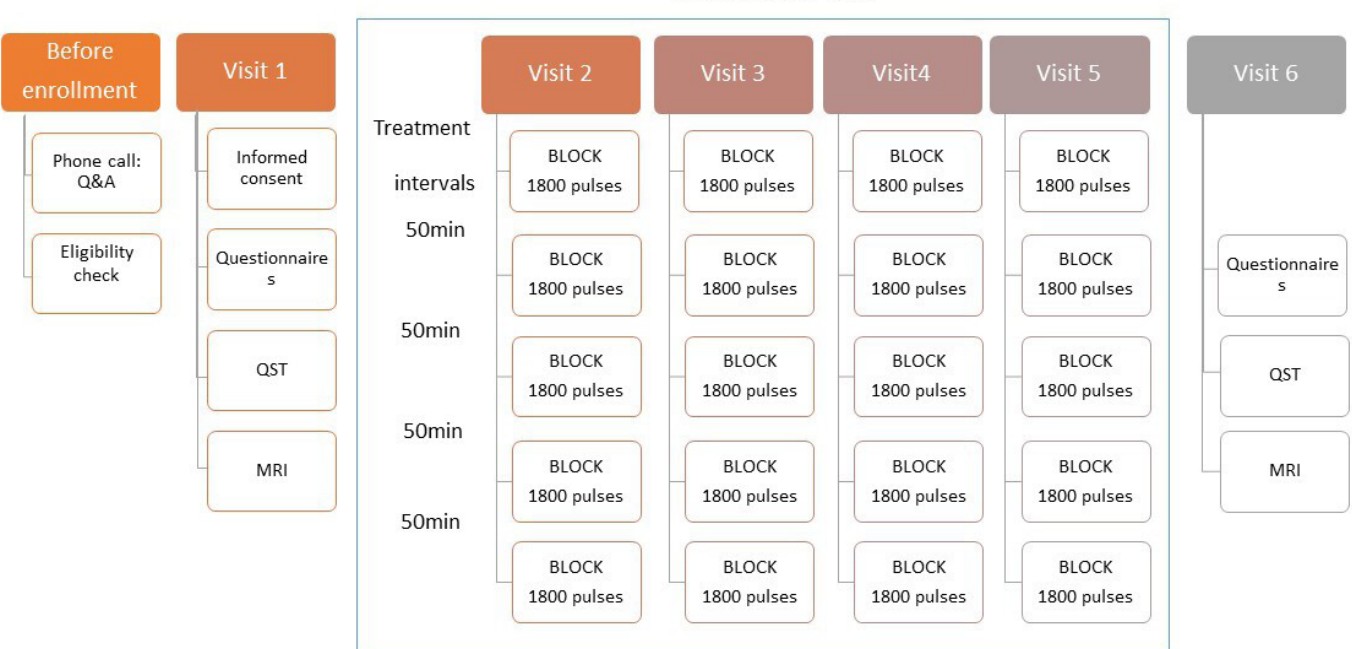

**Figure 1** Pilot study overview. iTBS, intermittent theta burst stimulation; QST, quantitative sensory testing.

participant then expresses interest in the new study and agrees for their contact details to be passed on to the new study team, the latter will contact them.

Participants may be initially invited to take part in the study by sending a short format participant information sheet (PIS) form that summarises the main points of the study for consideration of interest; participants who contact for further information will be sent the final PIS. Alternatively, they may find the short format PIS as a poster and contact us. Posters may be put around the Nottingham University Hospital Trust, the Medical School and other University of Nottingham sites, public sites such as public libraries and leisure/community centres. Posters may also be placed on local press, including but not limited to The Nottingham Post and Metro. The same advert may be featured on UK websites related to chronic pain, including but not limited to the Pain Centre Versus Arthritis webpages (http://www.nottingham.ac.uk/paincentre).

Participant enrolment and retention will be continually monitored, and numbers fed back at the study team meetings. Feedback from participants will be used to maintain recruitment and retention rates.

### Inclusion criteria

1. 18–75 years of age and able to consent.
2. Chronic knee pain (minimum 6 months duration) with visual analogue scale (VAS) score ≥4/10.
3. Able to accommodate the study visits without disruption to their jobs or other responsibilities and without experiencing excessive physical strain or other distress.

### Exclusion criteria

Patients are admitted to the study if they also meet the following criteria:

1. No contraindication to MRI and TMS.
2. No major medical or neurological conditions unrelated to the pain condition.
3. Not scheduled for total knee replacement surgery within 1 month of study visits or other considerable treatment change.
4. No change in pain medication in the past 4 weeks.
5. Not on centrally active medication other than stable antidepressants or on opioidergic analgesic treatment.
6. Not on medication, alcohol or recreational drugs, thought to increase risk of seizure and/or syncope.
7. Not experiencing frequent headaches.

Participants will be reimbursed for travel expenses and provided with free drinks and food during study sessions but will not be compensated for their time.

### Transcranial magnetic stimulation

TMS will be delivered using the Magstim Horizon Performance system with StimGuide Navigated TMS package (The Magstim Co Ltd). Single-pulse TMS will be applied to the left motor cortex using a handheld figure-of-eight coil (diameter 7 cm) to determine the RMT. RMT is defined as the minimum stimulator output intensity needed to achieve a minimum motor evoked potential-amplitude of 50 µV in the completely relaxed abductor pollicis brevis-muscle in at least 5 out of 10 trials. For all treatments, the iTBS protocol will be delivered using a fixed 7 cm diameter coil (E-z Cool Coil and support arm) consisting of three pulses (aimed at 80% RMT, but may be lowered to avoid participant discomfort) of 50 Hz repeated at 200 ms (5 Hz) intervals with a total 1800 pulses per block, 9000 pulses per visit/day and 36 000 pulses in total/treatment week.

The staff operating the TMS device (SZ, senior research nurse with 3 years experience in rTMS treatment/trials; DH senior research fellow with 2 years experience in research TMS) are trained on the MAGSTIM system and adhere to all safety instructions.

### MRI acquisition

Imaging data will be acquired on a GE Premier 3T MR scanner using a 48-channel head coil (GE Healthcare, USA). Structural scans include high-resolution T1-weighted (T1w) and T2-weighted (T2w) images for the purpose of image registration. The T1w images will be acquired using 3D MPRAGE sequence (voxel size=1 mm isotropic, FOV=256, matrix=256, 256 sagittal slices, TI=800 ms, FA=8°) and T2w will be images acquired using 3D Cube T2 FLAIR sequence (voxel size=1 mm isotropic, FOV=256, matrix=256, 256 sagittal slices, TR=6300 ms, TE=121 ms, TI=1787 ms). RsfMRI data will be acquired using whole-brain 2D GE-EPI sequence (TR=1400 ms, TE=35 ms, flip angle=68°, in-plane FOV=212×212 mm, 57 slices, slice thickness=2 mm, voxel size=2 mm isotropic, hyperband factor=3, ARC factor=2, 15 min total scan time). To measure the $b_0$ field for correction of echoplanar imaging (EPI) distortions, we will also acquire two SE-EPI images with reversed phase encoding directions. The SE-EPI images have the same geometry, echo spacing and phase encoding direction parameters as the GE fMRI scans.

### Image processing and TMS neuronavigation

Image pre-processing follows the same procedure as that described in Pszczolkowski et al.[10] In brief, the structural T1w images are brain extracted, bias corrected and segmented into white matter (WM), grey matter (GM) and cerebrospinal fluid probability maps which are binarised at a probability of 0.98 and then eroded using a spherical kernel with a radius of 2 voxels to obtain the final tissue labels. The blood oxygenation level dependent (BOLD) rsfMRI undergo EPI distortion correction, motion correction, slice timing correction, smoothing, denoising and high-pass filtering. Finally, the WM and GM tissue labels are transformed into rsfMRI BOLD space and used to regress out the WM and GM time series from the processed rsfMRI BOLD data.

For identification of the iTBS treatment target, a methodology similar to Pszczolkowski et al[10] will be followed. However, in the current pilot trial the DLPFC searchlight described in Fox et al[17] will be used as primary volume

of interest, and the background removal procedure is based on a threshold computed automatically rather than by visual inspection. This threshold is computed as the 98th percentile of the T1w intensities within a mask of background (air) voxels around the outside of the skull in MNI space which is previously transformed into T1w space.

### Quantitative sensory testing

Central pain inhibition will be tested pre-TBS and post-TBS treatment using the psychophysical paradigm of conditioned pain modulation (CPM). The test stimulus will be pressure pain threshold (PPT) and the conditioning stimulus ischaemic pain caused by a blood pressure cuff placed at the calf of the unaffected leg. The test stimulus (PPTs) will be administered at three body sites (on the affected side), including trapezius muscle, metacarpophalageal joint of the thumb and the patella. If no preference is shown with respect to the affected side, then the patients non-dominant body side will be examined. All QST assessment are completed by DH with >10 years experience.

CPM is calculated as follows:

CPM (%)=(PPTpost-conditioning–PPTpre-conditioning)/PPTpre-conditioning×100.

Preconditioning PPTs will be used to calculate SEM and >±2×SEM to determine CPM effects in response TBS treatment.

### Sham, randomisation and blinding

The sham intervention will be given through the same coil as the iTBS treatment and both active and sham will be administered over the left DLPFC using neuronavigation coordinates derived from rsfMRI. For the sham group, pulses will be delivered at 30% stimulator output by orienting the coil perpendicular to the scalp surface with direct contact. This will ensure that an appropriate audible noise is created to reinforce the integrity of the treatment blinding.

The study statistician will generate allocation by randomising blocks of participants, ensuring age and sex matching between arms. Block randomisation (block size 6) will be used and determined using online tools (http://www.jerrydallal.com/random/randomize.htm). Initially 30 subjects will be randomised with 3 subjects per group in each block. Given the recruitment rates and feasibility of recruiting all subjects within the time frame, the next subjects will be randomised similarly. However, if recruitment suggests a shortfall of subjects within the time frame the block allocation will be adjusted such that a higher rate of subjects are recruited to the treatment group such as to ensure sufficient numbers in that group. This is expected to be at most a small adjustment such that the block of 6 will still be sufficient to effectively mask assignment, for example, 4 into the treatment group and 2 into the sham group. The study coordinator (BM) will enrol and assign participants.

Study participants will be blinded to the intervention allocation and participants will be asked whether they can guess the type of intervention as part of a TMS exist interview/questionnaire. The study team involved in delivering the intervention cannot be blinded, but the data analysis team will be blinded to which intervention was delivered. Participants will be unblinded if they were to experience adverse events.

### Assessments

Data will be collected for the following

1. MRI at visit 1 and at visit 6 (pretreatment and post-treatment).
2. QST at visit 1 and at visit 6 (pretreatment and post-treatment).

REDCap (https://www.project-redcap.org/) will be used to administer online questionnaires:

Pain characteristics

1. A numerical pain VAS (0–10) as online daily pain report.
2. Weekly questionnaires:
   Intermittent and Constant Pain Score (ICOAP),[18] the Central Aspects of Pain in the Knee (CAP-Knee) questionnaire.[19]
3. At visit 1 and 6:
   painDETECT,[20] Pain Catastrophizing Scale (PCS),[21] ICOAP,[18] General Anxiety Depression scale (GAD-7),[22] Depression Anxiety Stress Scales (DASS-21),[23] Hospital Anxiety and Depression Rating-Scale (HADS),[24] Patient Health Questionnaire (PHQ-9),[25] CAP-Knee,[19] Montreal Cognitive Assessment test.[26]

### Outcome metrics and measures

Outcome metrics are changes between end of the intervention (visit 6) and baseline (visit 1) of optimised neuroimaging markers of the DPMS that will be established from independent datasets and predefined in the analysis plan and second markers of central pain modulation, derived from QST.

### Outcome measures

#### Primary

1. Variance of changes in pre-selected neuroimaging functional markers of the descending pain modulatory system at final versus baseline visit.
2. Variance of change in CPM (pain intensity change with/without conditioned pain stimulus) at final versus baseline visit.
3. Number of participants recruited and retained on the trial.
4. Number of iTBS sessions completed at 80% RMT.

#### Secondary

1. Variance of change in reported pain intensity, burden and catastrophising at final versus baseline study visit (ICOAP, PCS).
2. Change in negative affect (HADS) at final versus baseline study visit.

3. Variance in physiological and psychophysical changes (acute experimental pain) at final versus baseline study visit.
4. Variance of PROMs (daily online recording pain, well-being, sleep) over 16 weeks from treatment versus enrolment-baseline.

## Adherence, tolerability, withdrawal and pausing of study trial

Adherence in this study will be measured by participants' attendance at study visits, completion of all study measures and procedures, and TMS treatments. We will also record the tolerability of 80% RMT and any necessary adjustments of %RMT and NIBS stimulation due to discomfort. Safety instructions also include not being intoxicated by alcohol or non-prescription drugs at any study visit, and any history of heavy binges of alcohol or intermittent hypnotic or benzodiazepine use in the 24 hours before a treatment session because of these behaviours would increase the risk of seizures or syncope with TMS.

Participants who fail to comply with safety instructions can be withdrawn from the study at any time. Study withdrawal could be requested anytime by the participants.

Tolerability of the accelerated protocol will be individually monitored in the first 10 participants with planned meeting of the study team with the independent medical safety advisor (RT) for go/no-go decision-making should the study retention exceed expected levels for rTMS studies or based on concerning individual feedback. Within the first 10 participants we did not observe withdrawal due to protocol non-compliance.

## Known side effects and discomfort

The staff operating the TMS device (SH 3 years, DH 2 years) are trained and adhere to all safety instructions.

As it directly influences activity of nerve cells, theoretically there is the risk of a seizure although the risk level is considered low. The cumulative number of TMS pulses is 36 000 which is substantially lower than the accelerated Stanford protocol and lower than the FDA approved protocol (48 000–72 000 for 4–6 weeks).

Staff and participants will be wearing ear protectors because of the risk of tinnitus and hearing impairment from the loudness of the TMS.

The majority of TMS treatments are very well tolerated, occasionally however a short-lasting headache can occur. Depending on an individual's facial anatomy, facial nerves can be stimulated at the same time which would result in muscle twitches in the face; these however cease as the stimulation is stopped.

Should a participant experience any side effects and find it too uncomfortable, the treatment session can be stopped immediately but where possible, adjustments will be made to ease discomfort. To reduce participants' discomfort on the first day of the session, the power was individually ramped to 80% RMT or lower depending on subjective tolerability.

A standard operating procedure has been developed that defines the discontinuation of the trial intervention, referral to treatment as required and risk management of the most common serious side effect (seizure) and more common side effects like headache and dizziness.

## Safety assessment

Despite the safety report on accelerated iTBS protocols, side effects and tolerability in a chronic pain population were unknown, hence the trial will be monitored with the intention to stop should a larger than expected number of side effects be reported or low compliance be found. Within the first 10 participants we did not record any serious side effects. Any safety-related termination decision would be taken by the medical safety advisor.

## Feedback from public and patient involvement: a lay interest group for clinical research

Patients and healthy lay persons interested in clinical research were invited to give feedback on the study concept and separately the specific protocol. Participants of the first public and patient involvement (PPI) were positive about the possibility of a non-invasive treatment that may allow to replace medication. For the specific study protocol, PPI participants indicated that attending the TMS part with its up to 6 hours long visits for 4 days during a week would be a considerable burden for them, especially for those who suffer from pain-related partial disabilities. Other feedback criticised that this is only open for patients who are mobile, not actively working or on long term sick leave, and that a study of such dimension would need to include inconvenience allowances for participants. We hence included failure to recruit and protocol compliance in the feasibility and acceptability assessment.

## Sample size

The pilot aims to recruit n=45 (30 active, 15 sham iTBS) participants. In this pilot study, we estimate the variance of the outcome measures, and adjust for the upper confidence level.[27]

## Handling and storage of data

Hardcopies of consent, notes, and data will be stored securely in lockable closets. Digital data will be handled and stored safely on university-owned password-protected servers. Data will be entered in a REDCap database. To maintain data quality one person will enter the data and a second will check and validate the data. Data will be audited on a monthly basis and any issues raised at the biweekly study team meetings, one of whose functions is to monitor quality.

## Data analysis

Brain imaging changes will be assessed using current version of standardised and well documented image analysis such as the NIHR Nottingham BRC imaging pipeline (https://github.com/SPMIC-UoN) that has implemented and combined routines from the most suitable advanced image analysis packages, such as FSL,

MRConn and the specific protocol with statistical analysis plan will be documented before data analysis will occur.

## Trial status

The study started in July 2021 until July 2024. The recruitment opened 12 May 2022 and is anticipated to be completed by 31 December 2023. As of March 2023, 28 participants completed the trial.

Given the pilot nature and small size of the trial, no DMC was needed but instead an independent experienced neurologists (RT) acted as safety advisor to optimally manage the seizure risk.

## ETHICS AND DISSEMINATION

All participants will provide written informed consent for taking part in the study. Ethical approval has been obtained by the South West—Cornwall and Plymouth Research Ethics Committee (REC reference: 21/SW/0079). For the use of clinical facilities, approval has been obtained from Nottingham University Hospital NHS Trust (RAS 298509 Confirmation of Capacity and Capability).

Results of the feasibility and pilot trial will be submitted for publication in open access, peer-review journals, for presentation at scientific conferences and may be shared with participants and PPI/E advisors.

**Author affiliations**
[1]Mental Health and Clinical Neurosciences, School of Medicine, University of Nottingham, Nottingham, UK
[2]Nottingham NIHR Biomedical Research Centre, University of Nottingham, Nottingham, UK
[3]Pain Centre Versus Arthritis, University of Nottingham, Nottingham, UK
[4]Academic Rheumatology, School of Medicine, University of Nottingham, Nottingham, UK
[5]Clinical Neurology, Nottingham University Hospital Trusts, Nottingham, UK
[6]Adult Mental Health, Nottinghamshire Healthcare NHS Foundation Trust, Nottingham, UK

**Contributors** MD performed extensive literature review of TMS pain studies and connectivity analyses, suggested part of the questionnaires, obtained ethical approval and liaised with clinical and regulatory bodies for study set up, negotiated space within suitable facilities, prepared physical setup, purchased study necessities, assisted with training, and organised feedback from lay interest groups. DPA conceived the study, hypotheses, designed the study protocol and oversees the trial. DH designed the CPM assessment in this study, adapted MRI scanning protocols, advised on TMS and coil positioning, takes care of day-to-day study governance and edited the manuscript during rebuttal. SH contributes to the trial as research nurse. BM contributed to the recruitment process design, prepared the database and paper/electronic questionnaires and assisted with data collection. SL provided feedback on treatment protocol. RT contributed to the recruitment process review, safety medical review and suggested part of the questionnaires. SPP contributed in writing the software for imaging-based personalised connectivity-guided target-selection and manuscript preparation. CRT provided support with sample size estimations, advised on the statistical design and undertakes randomisation. RM advised on the trial design and clinical outcome metrics, and provided feedback on manuscript draft stages. DW provided funds for the trial through his grant from Versus Arthritis UK and provided feedback on the study design and draft manuscript before submission. MD and DPA prepared the manuscript and edited the manuscript during rebuttal. All coauthors provided feedback on the draft before submission.

**Funding** This trial is funded by Versus Arthritis UK and NIHR Nottingham BRC Nottingham. RT received support in part from the UK MRC (CARP MR/T024402/1). None of the funders have active roles in the design of the study, writing of the report, nor are involved in decisions regarding publications.

**Competing interests** None declared.

**Patient and public involvement** Patients and/or the public were not involved in the design, or conduct, or reporting, or dissemination plans of this research.

**Patient consent for publication** Not applicable.

**Provenance and peer review** Not commissioned; externally peer reviewed.

**ORCID iDs**
Marianne Drabek http://orcid.org/0009-0008-8976-9858
Christopher R Tench http://orcid.org/0000-0001-9067-0494
Richard Morriss http://orcid.org/0000-0003-2910-4121

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
