## [Reviewer comments · BMJ Open]

ARTICLE DETAILS

TITLE (PROVISIONAL)	Brain connectivity-guided, optimised theta burst transcranial magnetic stimulation to improve central pain modulation in knee osteoarthritis pain (BoostCPM): protocol of a pilot randomised clinical trial in a secondary care setting in the UK.
AUTHORS	Drabek, Marianne; Hodkinson, Duncan; Horvath, Suzanne; Millar, Bonnie; Pszczolkowski Parraguez, Stefan; Tench, Christopher; Tanasescu, Radu; Lankappa, Sudheer; Morriss, Richard; Walsh, David; Auer, Dorothee

VERSION 1 – REVIEW

REVIEWER	Chowdhury, Nahian Neuroscience Research Australia
REVIEW RETURNED	31-Mar-2023

GENERAL COMMENTS	The authors report a pilot RCT protocol for the use of optimized TBS to improve CPM symptoms of knee OA. This is a very important and exciting trial, particularly due to the use of individually optimized dLPFC target, and the intensity of the protocol. However, the protocol paper is lacking important details which will be helpful. I also advise the authors to go through the SPIRIT protocol to ensure all necessary items are reported (it would be helpful to actually report which page is located where). Here are my specific comments: Introduction - Be careful with statements such as “Best established method” -> many would disagree that TMS is the best approach to non-invasively change brain networks and restore connectivity patterns-The rationale for using rTMS for pain appears to be based on depression studies, but others would suggest rTMS modulates endogenous pain pathways directly rather than targeting the mood system -> the introduction would benefit from explaining the mechanism of action of rTMS from both perspectives- Page 4 – other quality concerns of rTMS studies are lack of blinding, details of the randomization sequence and inadequate sham conditions- The introduction needs more detail is required on the optimized target approach as this is a key novelty of the study – what does it mean and why might it be better- The rationale for using chronic knee pain patients is not well explained – why might knee pain patients benefit from optimized dLPFC stimulation specifically Method
--

	 - Page7 Line 13 Five blocks of iTBS – not iTBC - Page 7 line 13 Should be an interBLOCK interval of 50 minutes, not intersession - Page 7 :Line 53 – “were acquired” should be future tense - Page 7 TMS – what coil orientation will be used? - Page 8 Line 26 “is followed” – please have consistent future tense (or consistent tense in general) - Page 8 Line 28 – “Different here” can be replaced with “However in our study” - Then exactly will participants be debriefed about the presence of two conditions, and when exactly will they be asked to guess (at the end of the study?) - Blinding has always been an issue in rTMS studies, many researchers try to improve blinding using sham coils that produce the same click sound, or even using electrical stimulation to produce the flicking sensation. Tilting the active coil doesn’t seem effective enough and doesn’t address quality concerns of previous rTMS studies well enough. - Please detail how the randomization schedule will be generated and how this will be concealed from those who will be blinded - Are there situations where any researchers need to unblinded? E.g. unexpected adverse events - Outcome measures: it is unclear what is meant by “variance of changes” between baseline and visit - The recruitment/inclusion criteria should go at the start of the methods - Inclusion criteria – chronic knee pain (6 months) – most days than not? - The mood, anxiety, pain catastrophizing criteria is not clearly explained – what is the rationale for using “above population average” - It is ideal to have the criteria that start with “not” as exclusion criteria - Define “unstable” pain medication - Is past knee replacement or joint injection an exclusion criteria? - Not clear why there is uneven participants in active and sham groups
--	---

REVIEWER	Adhia, Divya University of Otago - Dunedin Campus, Surgical Sciences
REVIEW RETURNED	06-Apr-2023

GENERAL COMMENTS	Thank you for the opportunity to review this interesting study protocol. This is an interesting study. Please see below are some of my comments to improve the manuscript and provide more details on the procedures: Trial design (line 30): Why have authors chosen an allocation ratio of 2:1 compared to 1:1? Hypothesis 1: It will be good to add a direction to the hypothesis. Do the authors expect the activity and functional connectivity between the DLPFC and Right insula to increase ? p.7 line 32: Define ABP in its full form as it appears first time here p.8 line 42: CPM procedure: PPT will be applied to the unaffected leg, but what if the participants have OA in both knees? Would knee with less pain be tested then?
--

	p.9 line 15: I cannot find the coordinates of the L DLPFC that will be targeted. The authors need to justify why they choose L DLPFC specifically for people with Knee pain. Or is it more based on depression studies? p.9 lines 8 and 9: The CPM procedure in methods didn't mention when pain intensity will be measured? Feasibility measures (e.g., number of participant recruited/retained, number of treatment sessions completed) needs more specific details, how long will you advertise for? who will keep track of recruitment? Inclusion criteria: the cut off scores for the mood, anxiety and PCS scales need to be included. Include examples for contraindications to the MRI, TMS. What will be the exclusion criteria? Data analysis: This appears very minimal, and needs to be reported in details for all the primary, secondary and the mechanistic outcomes. General comments: The author uses a lot of abbreviations that are not defined and makes it hard to read. More details about outcome assessors and the treating therapists needs to be provided (e.g., qualifications, experience, etc) How will tolerability, acceptability, and safety measured? What will be the recruitment period? What are the start and end dates of the study? Discussion can be more refined to include previous studies of TMS in chronic pain.
--	---

VERSION 1 – AUTHOR RESPONSE

Reviewer: 1

Dr. Nahian Chowdhury, Neuroscience Research
Australia Comments to the Author:

The authors report a pilot RCT protocol for the use of optimized TBS to improve CPM symptoms of knee OA. This is a very important and exciting trial, particularly due to the use of individually optimized dLPFC target, and the intensity of the protocol.

We thank the reviewer for their interest in the study.

However, the protocol paper is lacking important details which will be helpful. I also advise the authors to go through the SPIRIT protocol to ensure all necessary items are reported (it would be helpful to actually report which page is located where).

Here are my specific comments:

Introduction

- Be careful with statements such as “Best established method” -> many would disagree that TMS is the best approach to non-invasively change brain networks and restore connectivity patterns

-The rationale for using rTMS for pain appears to be based on depression studies, but others would suggest rTMS modulates endogenous pain pathways directly rather than targeting the mood system

-

> the introduction would benefit from explaining the mechanism of action of rTMS from both perspectives

- Page 4 – other quality concerns of rTMS studies are lack of blinding, details of the randomization sequence and inadequate sham conditions

- The introduction needs more detail is required on the optimized target approach as this is a key novelty of the study – what does it mean and why might it be better

- The rationale for using chronic knee pain patients is not well explained – why might knee pain patients benefit from optimized dIPFC stimulation specifically

#Introduction comments: The controversial statement has been removed, the introduction has been edited in response to the helpful comments to clarify the trial motivating rationale.

Method

- Page 7 Line 13 Five blocks of iTBS – not iTBC Corrected
- Page 7 line 13 Should be an interBLOCK interval of 50 minutes, not intersession Corrected
- Page 7 :Line 53 – “were acquired” should be future tense Corrected
- Page 7 TMS – what coil orientation will be used?

Coil orientation is individually defined using the neuronavigation procedure described previously in Pszczolkowski et al. and referenced in the manuscript.

- Page 8 Line 26 “is followed” – please have consistent future tense (or consistent tense in general) Corrected.

- Page 8 Line 28 – “Different here” can be replaced with “However in our study” corrected.

- Then exactly will participants be debriefed about the presence of two conditions, and when exactly will they be asked to guess (at the end of the study?)

Patients will be assessed at their follow-up visit (final research visit at end of intervention) with an exit ‘TBS Experience Questionnaire’. The questionnaire includes the following question: *“Do you think you received active TBS or inactive (sham) TBS administration?” Active TBS | Inactive (sham) TBS | I don’t know/cannot estimate*

- Blinding has always been an issue in rTMS studies, many researchers try to improve blinding using sham coils that produce the same click sound, or even using electrical stimulation to produce the flicking sensation. Tilting the active coil doesn’t seem effective enough and doesn’t address quality concerns of previous rTMS studies well enough.

Sham coils are currently unavailable for the Magstim Horizon Performance system. In our experience, the chosen approach is effective for blinding in TMS naive participants. The study is a feasibility and pilot trial and the TBS experience questionnaire that we will collate will directly assess whether the sham / TBS allocation was effective or may have been compromised.

- Please detail how the randomization schedule will be generated and how this will be concealed from those who will be blinded.

There is no reason for stratification in this study. Block randomisation (block size 6) will be used determined using online tools (<http://www.jerrydallal.com/random/randomize.htm>). Initially 30 subjects will be randomised with 3 subjects per group in each block. Given the recruitment rates and feasibility of recruiting all subjects within the time frame, the next 30 subjects will be randomised similarly. However, if recruitment suggests a shortfall of subjects within the time frame the block allocation will be adjusted such that a higher rate of subjects are recruited to the treatment group such as to ensure sufficient numbers in that group. This is expected to be at most a small adjustment such that the block of 6 will still be sufficient to effectively mask assignment, for example 4 into the treatment group and 2 into the sham group.

Only participants not TMS operators are blinded. Blinding for data analysis will be achieved excluding allocation information from primary data and metadata used for the analysis.

- Are there situations where any researchers need to unblinded? E.g. unexpected adverse events

The research team applying the TMS intervention cannot be blinded for practical reasons, hence will not need unblinding in the event of adverse events.

Please see comments above.

- Outcome measures: it is unclear what is meant by “variance of changes” between baseline and visit
- The recruitment/inclusion criteria should go at the start of the methods Corrected.

- It is ideal to have the criteria that s- Inclusion criteria – chronic knee pain (6 months) – most days than not?

Yes, constant knee pain refers to reported pain most days for at 6 months or longer.

Participants have constant knee pain (most days) for greater or equal to 6 months; we clarified this in the manuscript.

- The mood, anxiety, pain catastrophizing criteria is not clearly explained – what is the rationale for using “above population average”

This section is no longer relevant. We initially aimed to enrich the study population for negative affect based on above population scores for one of three questionnaires probing affect (pain catastrophising, mood and anxiety) as a pragmatic approach to balance enrichment versus feasibility. Yet, at the initial set-up investigator meeting it was felt that the planned screening questionnaires would not allow a robust enrichment enrichment, and alternate questionnaires were proposed requiring Ethics amendment. Due to the time constraints of the trial and remaining uncertainty about effectiveness of the revised approach we decided to drop this inclusion criterion and instead report the affect profile of the recruited study population in comparison to published age- and sex matched populations and the larger research-ready community based cohort 'Investigating Musculoskeletal Health and Wellbeing' for complete phenotyping of the cohort's affect comorbidity and detection of selection bias.'

start with “not” as exclusion criteria Corrected.

- Define “unstable” pain medication Wording improved.
- Is past knee replacement or joint injection an exclusion criteria? Not unless in the past four weeks

Not clear why there is uneven participants in active and sham groups

This was a pragmatic compromise. Here we plan a pilot study to estimate an effect size to use for a sample size calculation in a subsequent study. We use the non-central t-distribution method (1) suggesting around 30 subjects per group is optimal (2). To ensure blinding while minimising the number of participants recruited, an allocation ratio of 2:1 (active/sham) is chosen to prioritise good quality estimates for the intervention arm. This totals to N=45. Based on our experience we will aim to recruit 50 participants to compensate 10% drop-out.

Julious SA, Owen RJ. Sample size calculations for clinical studies allowing for uncertainty about the variance. *Pharm Stat.* 2006 Jan-Mar;5(1):29-37. doi: 10.1002/pst.197. PMID: 17080926.

Whitehead AL, Julious SA, Cooper CL, Campbell MJ. Estimating the sample size for a pilot randomised trial to minimise the overall trial sample size for the external pilot and main trial for a continuous outcome variable. *Stat Methods Med Res.* 2016 Jun;25(3):1057-73. doi:

10.1177/0962280215588241. Epub 2015 Jun 19. PMID: 26092476; PMCID: PMC4876429

Reviewer: 2

Dr. Divya Adhia, University of Otago - Dunedin
Campus Comments to the Author:

Thank you for the opportunity to review this interesting study protocol. This is an interesting study. Please see below are some of my comments to improvise the manuscript and provide more details on the procedures:

We thank the reviewer for their interest in our study.

Trial design (line 30): Why have authors chosen an allocation ratio of 2:1 compared to 1:1?

This was a pragmatic compromise. Here we plan a pilot study to estimate an effect size to use for a sample size calculation in a subsequent study. We use the non-central t-distribution method (1) suggesting around 30 subjects per group is optimal (2). To ensure blinding while minimising the number of participants recruited, an allocation ratio of 2:1 (active/sham) is chosen to prioritise good quality estimates for the intervention arm. This totals to N=45. Based on our experience we will aim to recruit 50 participants to compensate 10% drop-out.

Julious SA, Owen RJ. Sample size calculations for clinical studies allowing for uncertainty about the variance. *Pharm Stat.* 2006 Jan-Mar;5(1):29-37. doi: 10.1002/pst.197. PMID: 17080926.
Whitehead AL, Julious SA, Cooper CL, Campbell MJ. Estimating the sample size for a pilot randomised trial to minimise the overall trial sample size for the external pilot and main trial for a continuous outcome variable. *Stat Methods Med Res.* 2016 Jun;25(3):1057-73. doi:

10.1177/0962280215588241. Epub 2015 Jun 19. PMID: 26092476; PMCID: PMC4876429

Hypothesis 1: It will be good to add a direction to the hypothesis. Do the authors expect the activity and functional connectivity between the DLPFC and Right insula to increase ?

Network hub and interaction changes are expected to be complex with insufficient data to justify directional regional hypotheses specifically at this stage. The most plausible assumption that we have made is that beneficial iTBS induced neuromodulation would 'normalise' the altered pain connectome and this has been added in the revision with an example of the key feature of enhanced anticorrelation between rAI-post DMN. As we prepare the final statistical analysis that will be documented on a public UoN repository before any trial data will be analysed, we will establish the key dysconnectivity hallmarks based on larger available local and public MRI datasets that include healthy controls arm to inform the specific directional hypotheses to be tested.

p.7 line 32: Define ABP in its full form as it appears first time here

This has been defined in full: "*abductor pollicis brevis (APB)-muscle*"

p.8 line 42: CPM procedure: PPT will be applied to the unaffected leg, but what if the participants have OA in both knees? Would knee with less pain be tested then?

To clarify, we state in the methods that: "The test stimulus (PPTs) will be administered at three body sites (on the affected side), including trapezius muscle, metacarpophalangeal joint of the thumb, and the patella." If no preference is shown with respect to the affected side, then the patient non-dominant body side will be examined. We include this additional information in the methods section.

p.9 line 15: I cannot find the coordinates of the L DLPFC that will be targeted. The authors need to justify why they choose L DLPFC specifically for people with Knee pain. Or is it more based on depression studies?

Targeting is individually defined using the neuronavigation procedure described previously in Pszczolkowski et al (13). This approach was optimised for modulation of the DLPFC-right anterior insula circuit as relevant for both treatment-resistant depression and chronic pain.

In response to your and the other reviewer's comments we reworded the trial motivation section in the introduction explaining that both from a direct pain modulatory perspective and the available evidence that negative affect augments pain progression, choice of the left DLPFC is well justified.

p.9 lines 8 and 9: The CPM procedure in methods didn't mention when pain intensity will be measured?

CPM testing will be performed pre- and post-TBS treatment. This section has been edited to include procedure timings.

Feasibility measures (e.g., number of participant recruited/retained, number of treatment sessions completed) needs more specific details, how long will you advertise for? who will keep track of recruitment?

Participant enrolment and retention will be continually monitored by the study co-ordinator, and numbers fed back to the whole study team and discussed at regular trial meetings. Feedback from participants will be used to maintain recruitment and retention rates.

Inclusion criteria: the cut off scores for the mood, anxiety and PCS scales need to be included.

This section is no longer relevant. We initially aimed to enrich the study population for negative affect based on above population scores for one of three questionnaires probing affect (pain catastrophising, mood and anxiety) as a pragmatic approach to balance enrichment versus feasibility. Yet, at the initial set-up investigator meeting it was felt that the planned screening questionnaires would not allow a robust enrichment enrichment, and alternate questionnaires were proposed requiring Ethics amendment. Due to the time constraints of the trial and remaining uncertainty about effectiveness of the revised approach we decided to drop this inclusion criterion and instead report the affect profile of the recruited study population in comparison to published age- and sex matched populations and the larger research-ready community based cohort 'Investigating Musculoskeletal Health and Wellbeing' for complete phenotyping of the cohort's affect comorbidity and detection of selection bias.'

Include examples for contraindications to the MRI, TMS. For MRI, this includes but is not limited to metal within the body but there can be safe exceptions (e.g. depending on location, metal used) and hence the safety assessment is done on an individual basis by an experienced MRI radiographer. For TMS, most safety criteria are listed (e.g. frequent headaches, medication).

What will be the exclusion criteria? Added to the respective section.

Data analysis: This appears very minimal, and needs to be reported in details for all the primary, secondary and the mechanistic outcomes.

The section has been updated, but not with full details as this will be part of a separate prospective data analysis plan with full details that will be published on appropriate repository before data analysis will commence.

General comments:

The author uses a lot of abbreviations that are not defined and makes it hard to read.

Abbreviations have been checked and defined.

More details about outcome assessors and the treating therapists needs to be provided (e.g., qualifications, experience, etc)

This has been added.

'The staff operating the TMS device (SZ, senior research nurse with 3 years' experience in rTMS treatment/trials; DH senior research fellow with 2 years' experience in research TMS) are trained on the MAGSTIM system and adhere to all safety instructions. '

For QST assessment, DH is the sole assessor with >10 years' experience.

How will tolerability, acceptability, and safety measured?

By study retention, tolerance of the intended motor threshold, observed and reported side effects and most importantly via participant feedback including an exit interview after the last intervention.

What will be the recruitment period? 12/05/2022-31/12/2023

What are the start and end dates of the study? July 2021- July 2024

Discussion can be more refined to include previous studies of TMS in chronic pain.

As requested by the Editor the Discussion section has been entirely removed.

VERSION 2 – REVIEW

REVIEWER	Chowdhury, Nahian Neuroscience Research Australia
REVIEW RETURNED	01-Aug-2023
GENERAL COMMENTS	The authors have adequately addressed my comments. I have no further queries.